# The nature and nurture of network evolution

Bin Zhou [1], Petter Holme[2,3], Zaiwu Gong [1], Choujun Zhan[4], Yao Huang [5], Xin Lu [6] & Xiangyi Meng [7,8] ✉

Although the origin of the fat-tail characteristic of the degree distribution in complex networks has been extensively researched, the underlying cause of the degree distribution characteristic across the complete range of degrees remains obscure. Here, we propose an evolution model that incorporates only two factors: the node's weight, reflecting its innate attractiveness (nature), and the node's degree, reflecting the external influences (nurture). The proposed model provides a good fit for degree distributions and degree ratio distributions of numerous real-world networks and reproduces their evolution processes. Our results indicate that the nurture factor plays a dominant role in the evolution of social networks. In contrast, the nature factor plays a dominant role in the evolution of non-social networks, suggesting that whether nodes are people determines the dominant factor influencing the evolution of real-world networks.

Pioneered by Helen Jennings in the 1930's[1], the degree distribution is a key characteristic of empirical network studies. Previous studies on the degree distribution of complex networks have primarily focused on the tail of the distribution, in particular when it exhibits a power law, which has led to the theory that "scale-free networks" are ubiquitous in nature[2–7]. Numerous network evolution models have been proposed to explain the mechanism that causes the fat-tail of the degree distribution to follow a power law[8–12], with the preferential attachment mechanism in the Barabási–Albert (BA) model being the most famous[13]. However, the debate about whether complex networks truly have scale-free properties has persisted[12,14,15]. Some scholars have proposed that we need to understand the scale-free properties and evolutionary origins of complex networks from a new perspective[16–18]. While the tail of the degree distribution may be approximated as a power law for many real-world networks[19–25], the bulk (the small-degree end) tends to bend off in various networks such as Facebook (friendships network), Google (informational network), the patent network of USA (technological network), etc[17,26]. In this work, we will propose a model for network evolution with an emergent degree distribution that fits observations throughout the degree range, including both the tail and the bulk.

Our proposed network evolution model incorporates only two parameters: an intrinsic node weight (a.k.a. "fitness"[27] or "quality"[28]) and the accumulated degree. These parameters effectively capture the dual influences of inherent characteristics (referred to as the "nature" factor) and environmental influences (referred to as the "nurture" factor) on the evolution of each node. We begin by demonstrating the core idea and formulation of our model. Following this, we proceed to solve the model analytically, focusing on deriving the analytical solutions for the distributions of degree $k$ as well as the degree ratio $\eta$, more commonly referred to as the degree–degree distance[17,29]. We find that the statistically optimal fit to these analytical solutions accurately reproduces both the degree distribution and degree ratio distributions of 32 real-world networks. Additionally, we verify that our model can produce the actual growth process in several networks. We find that the nurture factor of nodes predominantly influences the evolution of social networks, implying that the node degree has a greater impact on node evolution than the node weight in social networks. Whereas the nature factor of nodes plays a leading role in the evolution of non-

[1]Collaborative Innovation Center on Forecast and Evaluation of Meteorological Disasters, the Research Institute for Risk Governance and Emergency Decision-Making, School of Management Science and Engineering, Nanjing University of Information Science and Technology, 210044 Nanjing, Jiangsu, China. [2]Department of Computer Science, Aalto University, FI-00076 Aalto, Finland. [3]Center for Computational Social Science, Kobe University, Kobe, Hyogo 657-8501, Japan. [4]School of Computer, South China Normal University, 510631 Guangzhou, Guangdong, China. [5]School of Electrical and Computer Engineering, Nanfang College Guangzhou, 510970 Guangzhou, Guangdong, China. [6]College of Systems Engineering, National University of Defense Technology, 410073 Changsha, Hunan, China. [7]Network Science Institute and Department of Physics, Northeastern University, Boston, MA 02115, USA. [8]Department of Physics and Astronomy, Northwestern University, Evanston, IL 60208, USA. ✉e-mail: x.meng@neu.edu

social networks, suggesting that the impact of node weight on node evolution is greater than that of node degree in non-social networks. This observation implies that whether nodes are people plays a crucial role in determining the dominant factor driving the evolution of real-world networks. It also indicates that collective human behaviors, within the context of social interactions, tend to favor the nurture factor over the nature factor.

Not only does the model provide statistically optimal fittings to the observed distributions, but it also reveals the evolutionary origin of complex networks in terms of the interplay between both nature and nurture factors. Compared with the classical complex network models[8,13,28,30], the model still includes the preferential attachment mechanism, leading us to conclude that the scale-free property of complex networks should be understood as a mechanism, such as the preferential attachment mechanism, rather than a specific index, thus potentially resolving the long-standing debate about whether complex networks have scale-free properties.

## Results

### Nature–nurture model

Our study posits that the evolution of complex networks is closely tied to the interplay between two key factors: the node's weight reflecting its appeal within the network, which reflects the nature aspect of development, and the node's degree, which signifies its nurture factor. This coupling of the nature and nurture factors of nodes plays a crucial role in shaping the network's evolution, as shown in Fig. 1.

Before nodes join the network, we consider that their innate attractiveness are different in real world, similar to the Matthew effect[31]. For example, on Facebook, a user's social prestige, status, and influence serve as their innate weight, with most users being ordinary and only a few having high social prestige, status, and influence. The more social prestige, status, and influence an individual has, the more attractive they are[32]. In general, we assume that the distribution of node weight $\omega$ in a complex network follows a power-law distribution ~$\omega^{-\alpha}$, with $\alpha \geq 0$. This assumption also covers cases of a uniform distribution when $\alpha \to 0$, or a short-tail (e.g., exponential) distribution when $\alpha \to \infty$ (see Supplementary Discussion). The larger the nature

weight $\omega$ of a node, the higher its probability $\Pi(\omega)$ of establishing new links with other nodes.

On the other hand, the node's degree $k$ reflects its nurtured attractiveness, which is akin to the snowball effect[33] or recommendation systems[28]. Taking Twitter as an example, as a user's number of followers grows, their attractiveness to other users increases, further boosting their follower count. The larger the nurture degree $k$ of a node, the higher its probability $\Pi(k)$ of establishing new links with other nodes.

Consequently, nodes with larger $\omega$ and $k$ are more prone to establishing new links. This motivates us to choose the probability of a node being preferentially selected to form new links with other node as $\Pi(\omega, k) \sim \omega k +$ a positive constant. This formulation is in line with the approach taken in the Bianconi–Barabási model[30], where the probability is also a function of the product of $\omega$ and $k$.

Finally, we incorporate a cutoff parameter, $\omega_{max}$, which restricts the value of $\omega$ to fall within the range of 1 to $\omega_{max}$. This critical parameter serves to regulate the model's inclination towards either "nature" or "nurture." A smaller $\omega_{max}$ results in less variability in the distribution of the "nature" influence, suggesting that the model leans towards "nurture." Conversely, a larger $\omega_{max}$ allows for greater variability, indicating that the model favors "nature."

Taken together, our model is built on the following rules:

1. Initially, there are $N$ nodes but no link in the network. Each node $i = 1, 2, \cdots, N$ is assigned a weight $\omega_i$. Similar to other models with node weights generating the degree distribution[8,30], the weight for each node is randomly sampled from a truncated power-law probability distribution, following the form ~$\omega^{-\alpha}$, within a finite domain of $\omega \in [1, \omega_{max}]$.

2. At each time step, two nodes are randomly and independently chosen, and a link is established between them. The probability of choosing a node depends on the nature weight $\omega$ and the nurture degree $k$ of the node, given by

$$\Pi(\omega, k) = \frac{\omega(k + b)}{\sum_{i=1}^{N} \omega_i (k_i + b)}, \quad (1)$$

where $b$ is a positive constant.

3. After $T$ time steps, a network of $N$ nodes and $T$ links is generated.

The degree distribution of the nature-nurture model can be written as follows:

$$P(k) = \int_1^{\omega_{max}} d\omega_i c \omega_i^{-\alpha} n_i(T, k), \quad (2)$$

where $n_i(T, k)$ is the probability that node $i$ (of weight $\omega_i$) has degree $k$ at time step $T$ and $c = (1 - \alpha)/(\omega_{max}^{1-\alpha} - 1)$ is the normalization coefficient. Further approximations allow us to derive an analytical form of $P(k)$ (see "Methods").

To demonstrate that the model can accurately replicate multiple topological features, not just degrees, in complex networks, we examine a link-based characteristic: the degree ratio $\eta$, defined as the ratio of the larger and smaller degree for each link $(i, j)$, expressed as $\eta = \max(k_i/k_j, k_j/k_i)$. Note that this can also be reformulated as $\ln \eta = |\ln k_i - \ln k_j|$, which serves as a semi-metric on the set of edges[34]. Hence, $\eta$ (or more precisely, $\ln \eta$) is often referred to as the degree–degree distance[17]. Notably, in many empirical networks, the degree ratio distribution exhibits a clearer power-law behavior than the degree distribution, signifying its usefulness in examining the scale-free properties of networks. The degree ratio distribution is

**Fig. 1 | Nature versus nurture in network evolution.** Suppose there are two nodes of different node weights and degrees in the network. The red node has a larger weight but a smaller degree (with three incident links). The blue node has a smaller weight but a larger degree (with six incident links). As new links are added to the network, if the network evolution is nature-dominant, then new links prefer connecting to the red node; else, if the network evolution is nurture-dominant, then new links prefer connecting to the blue node.

given by

$$P(\eta) = \int_1^\infty \frac{2k_i N^2}{T} dk_i \int_1^{\omega_{max}} d\omega_i \int_1^{\omega_{max}} d\omega_j$$
$$c\omega_i^{-\alpha} c\omega_j^{-\alpha} n_{ij}(T, k_i, \eta k_i, (i,j)), \tag{3}$$

where $n_{ij}(T, k_i, k_j, (i,j))$ denotes the joint probability that nodes $i$ and $j$ have degrees $k_i$ and $k_j$ and they are also connected by a link (see "Methods").

The reliability of the analytical solutions of our model is demonstrated through a comparison of the degree distributions and degree ratio distributions obtained from simulations and Eqs. (2) and (3)), respectively (Supplementary Fig. 1). The agreement between the simulation and analysis results confirms the reliability of the analytical solutions of the nature–nurture model.

We also present two supplementary models to serve as controls: a nature-only model and a nurture-only model (see "Methods"). In the former, we eliminate the effect of the degree $k$, so that $\Pi(\omega, k) \to \Pi(\omega) \propto \omega$, only depending on $\omega$. The resulting degree and degree-ratio distributions are as follows:

$$P(k) \simeq \int_1^{\omega_{max}} c\omega_i^{-\alpha} \frac{1}{\sqrt{2\pi}\sigma_i} \exp\left[-\frac{(k-\mu_i)^2}{2\sigma_i^2}\right] d\omega_i, \tag{4}$$

which can be further approximated to a classical power-law distribution $P(k) \propto k^{-\alpha}$ [17], and

$$P(\eta) \simeq \int_1^{\omega_{max}} \int_1^{\omega_{max}} \frac{2c\omega_i^{-\alpha} c\omega_j^{-\alpha} d\omega_i d\omega_j}{T/N^2} \left(\frac{\eta\mu_j\sigma_i^2 + \mu_i\sigma_j^2}{\eta^2\sigma_i^2 + \sigma_j^2}\right)$$
$$\frac{\mu_i\mu_j/4T}{\sqrt{2\pi}\sqrt{\eta^2\sigma_i^2+\sigma_j^2}} \exp\left[-\frac{(\eta\mu_i-\mu_j)^2}{2(\eta^2\sigma_i^2+\sigma_j^2)}\right], \tag{5}$$

where $\mu_i = 2\omega_i T/N\bar{\omega}$ and $\sigma_i = (1-2\omega_i/N\bar{\omega})2\omega_i T/N\bar{\omega}$, and $\bar{\omega} = (2-\alpha)^{-1}(\omega_{max}^{1-\alpha}-1)^{-1}(1-\alpha)(\omega_{max}^{2-\alpha}-1)$ is the average node weight of the network.

In the nurture-only model, we eliminate the effect of the weight $\omega$, so that $\Pi(\omega, k) \to \Pi(k) \propto k + b$, only depending on $k$. In the $b \to 0$ limit, the resulting degree and degree-ratio distributions are [29]:

$$P(k) \simeq e^{-Bk} k^{A-1} B^A / \Gamma(A), \tag{6}$$

which is a power-law distribution with an exponential cutoff, and

$$P(\eta) \simeq 2B^{2A+2}\eta^A E_{-2A-1}((\eta+1)B)/\Gamma(A+1)^2, \tag{7}$$

where $A = b$ and $B = 2^{-1}bNT^{-1}$, respectively.

## Validation

We have gathered thirty-two real-world networks that span across social, informational, technological, biological and economic domains from the Colorado Index of Complex Networks (ICON). These networks vary in size, ranging from tens of thousands to hundreds of millions of nodes. Our data includes the most representative network platforms such as Facebook, Twitter, Wikipedia, Amazon, YouTube, Google, and Academia, among others. Descriptions for these networks can be found in Supplementary Table 1.

Figure 2 (and Supplementary Fig. 2) shows the optimal fitting results of the distributions of both degree $k$ [Eq. (2)] and degree-ratio $\eta$ [Eq. (3)] for thirty-two real-world networks. The parameters $N$ and $T$ in Eqs. (2) and (3) are fixed as the numbers of nodes and links of the fitted data, respectively. The optimal values of the fitting parameters $\omega_{max}$, $\alpha$, and $b$ are provided in Supplementary Table 2. We find that the nature-nurture model simultaneously reproduces both the degree and the

degree ratio distributions of real-world networks fairly well. These results suggest that the coupling of both nature and nurture factors of nodes plays an essential role in the evolution of complex networks.

In particular, Fig. 3A shows the optimal values of $\omega_{max}$ for the real-world networks, with blue and red circles representing eleven social and 21 non-social networks, respectively. We observe that the social and non-social networks are distributed in two distinct regions. In social networks, $\omega_{max}$ tends to be smaller, while in non-social networks, $\omega_{max}$ tends to be larger. This suggests that the nature factor of nodes plays a dominant role in the evolution of social networks, while the nurture factor of nodes plays a dominant role in the evolution of non-social networks. To corroborate this observation, we calculated the corrected Akaike Information Criterion (AICc)[35]—a statistical estimator that deals with the risks of both overfitting and underfitting—for the optimal fits of the distributions of $k$ and $\eta$ (Supplementary Table 3 and Fig. 3). This was conducted for the nature-nurture model as well as the control models, namely, the nature-only and nurture-only models. We find that the nature–nurture model is the most favored by AICc for 31 (96.9%) of the 32 real-world networks. By comparing only the control models [Fig. 3B, C], we find that the nurture-only model is favored by AICc for 81.8% of the social networks, yet the nature-only model is favored for 85.7% of the non-social networks. These results provide evidence that while the nature and nurture factors tend to dominate in the evolution of non-social and social networks, respectively, it is essential to consider the contributions from both aspects for an faithful representation of real-world network evolution.

Two networks, Academia (tracking citations between academic papers) and Zhihu (a Chinese Q&A forum), are accompanied by time-stamps, allowing us to explore how their degree and degree-ratio distributions evolve over various time periods. Figure 4 shows the fitting results for the initial, middle, and final stages during the evolution of the two networks. Again, the parameters $N$ and $T$ in Eqs. (2) and (3) are fixed as the numbers of nodes and links of the fitted data. The other fitting parameters, $\omega_{max}$, $\alpha$, and $b$, at each stage are fixed to the optimal values of the Academia and Zhihu networks (obtained from Supplementary Table 2).

The evolution fitting results (Supplementary Table 4) demonstrate that the nature-nurture model continues to simultaneously reproduce the distributions of $k$ and $\eta$ throughout the evolution process, from the initial to the final stage (Fig. 4). This confirms the model's ability to capture the evolutionary dynamics of complex networks. Moreover, the nature–nurture model continues to be the most favored by AICc, compared to the control models (Supplementary Table 5) during the evolution. Between the nature-only and nurture-only models, the former is more favored by AICc for the non-social network Academia, while the latter is more favored for the social network Zhihu. The consistency of results across static and evolutionary networks highlights the universal applicability of the nature-nurture model.

The biggest difference between nodes in social networks and those in non-social networks is that nodes in social networks represent users, who are people with strong subjectivity and self-modification abilities in the postnatal evolution, albeit limited by innate factors. On the other hand, nodes in non-social networks represent non-people entities with innate attributes and functions, generally lacking the ability to self-modify in the postnatal evolution process. In human society, we may believe that a person's efforts should carry more weight than their social background in determining social position. Therefore, in the evolution of social networks, it makes sense that the nurture factor plays a primary role. The result reveals a fresh aspect of the "nature vs. nurture" discussion from the perspective of network science: although both the nature and nurture factors impact individual human behaviors, the nurture factor assumes a more prominent role when determining collective behaviors within social networks, rather than focusing solely on individuals. Other systems have less

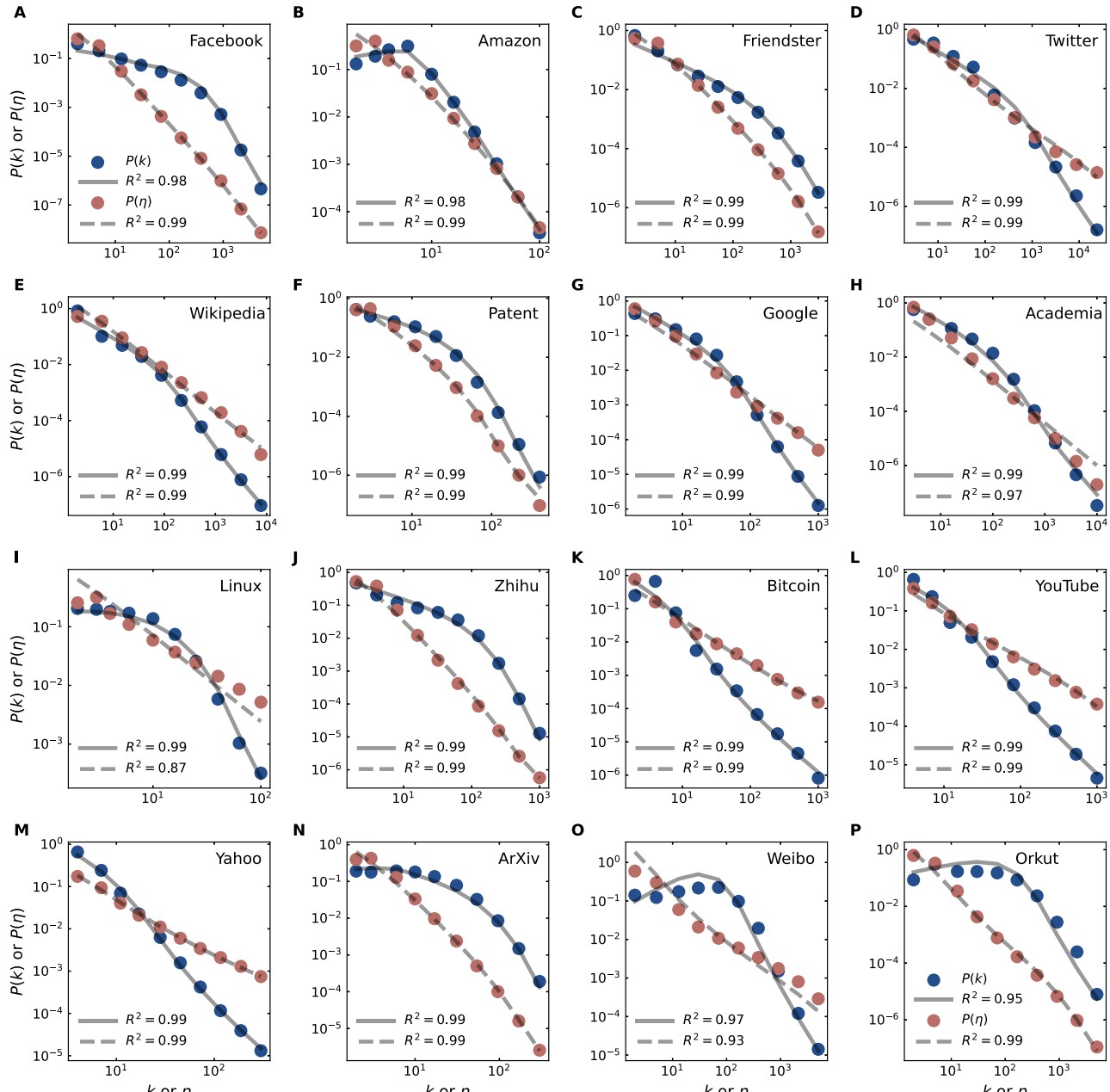

**Fig. 2 | Nature-nurture model fitting of real-world networks. A–P** The observed degree distribution $P(k)$ (blue) and degree-ratio distribution $P(\eta)$ (red) in 32 real-world networks (other 16 in Supplementary Fig. 2) are fitted based on Eqs. (2) and (3) of the nature−nurture model. The parameters $N$ and $T$ match the number of nodes and links in the empirical data. The other fitting parameters, $\omega_{max}$, $\alpha$, and $b$ are provided in Supplementary Table 2.

pronounced structural feedback, and are thus determined by the innate attributes to a larger extent. As such, we propose that in the evolution of non-social networks, the nature factor of nodes should play a leading role. Therefore, whether nodes in complex networks are people or not determines the dominant factor influencing the evolution of complex networks.

## Discussion

Since the publication of Galton's renowned paper in 1865[36], the exploration of the relative effects of nature (genetics) and nurture (environment) on individuals has remained a central focus in the fields of biology and sociology, leading to a vast body of literature on this subject[37–43]. However, there have been relatively few studies that approach this discourse from the perspective of complex systems. Our research provides a refreshing insight into the ongoing "nature vs.

nurture" discussion: while individual variations are significant (and may not be predictable), collective behaviors demonstrate predictability and can be categorized as either pro-nature or pro-nurture. This discovery underscores the potential of interdisciplinary studies that apply complex networks to diverse disciplines.

In conclusion, we propose a model of network evolution aiming to shed light on the evolutionary origin of complex networks. The optimal fitting results of the analytical solutions in the model reproduce the degree distributions and degree ratio distributions of both static and dynamic networks. These findings indicate that the coupling of both nature and nurture factors of nodes plays a crucial role in the evolution of complex networks, and our model can rather universally account for the evolution of complex networks. However, the strength of the nature and nurture components of the growth might vary, which furthermore gives a characterization of the network growth. In social

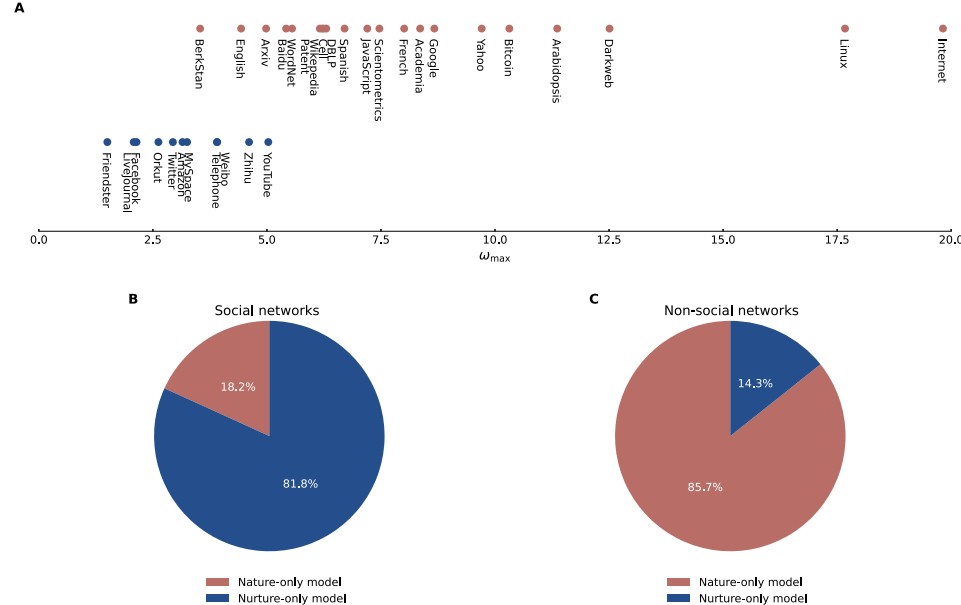

**Fig. 3 | Nature versus nurture in real-world networks. A** Optimal fitting parameter $\omega_{max}$ of the nature-nurture model in various real-world networks. Social networks (blue) generally exhibit lower $\omega_{max}$ values compared to non-social networks (red). **B**, **C** Preference for the nature-only (red) or nurture-only (blue) model in fitting social and non-social networks, respectively, based on the corrected Akaike information criterion for small sample sizes (AICc) provided in Supplementary Table 3.

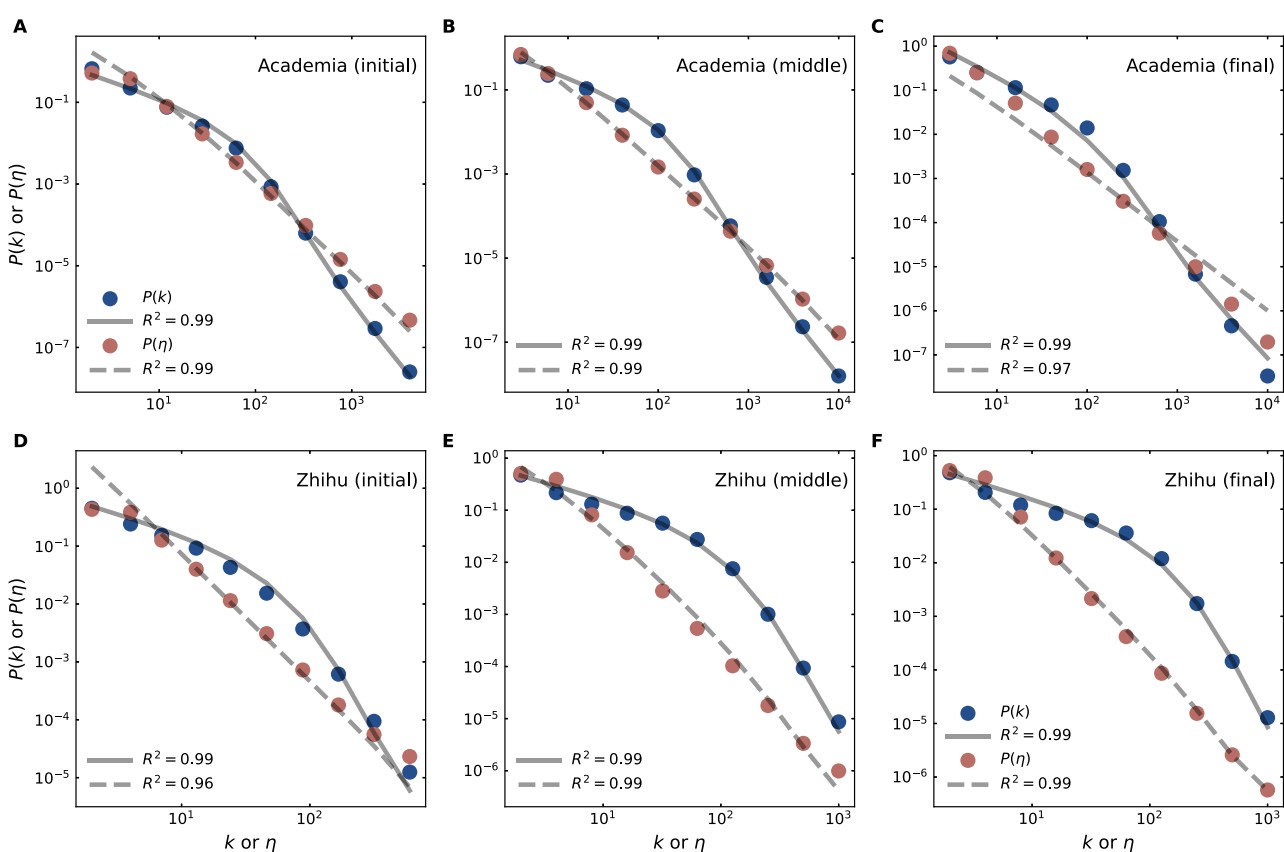

**Fig. 4 | Nature-nurture model fitting across network evolution.** The observed degree distribution $P(k)$ (blue) and degree-ratio distribution $P(\eta)$ (red) in **A**–**C** Academia (a non-social network) and **D**–**F** Zhihu (a social network) are captured at different timestamped stages. Solid and dashed lines represent fits based on Eqs. (2) and (3) of the nature–nurture model, with AICc provided (Supplementary Table 5). The parameters $N$ and $T$ match the number of nodes and links in the empirical data at each evolutionary stage (Supplementary Table 4). The other fitting parameters, $\omega_{max}$, $\alpha$, and $b$ are set according to the optimal values obtained in Supplementary Table 2.

networks, the nurture factor of nodes is dominant, implying that individuals can improve their social value through their acquired efforts instead of solely relying on their innate background. Conversely, in non-social networks, the nature factor of nodes plays a leading role, where the innate attributes and functions of agents provided by the system determine their acquired state and development in the system, suggesting that whether nodes are people determines the dominant factor influencing the evolution of complex networks.

In our work, we have not explicitly addressed the issue of network directionality. The primary goal of our study is to investigate the universal mechanisms that can be adaptable to the evolution of both undirected and directed networks. For directed networks, we treat the sum of node outdegrees and indegrees as the total degrees of a node, followed by calculating the degree distribution without explicitly delving into the directionality consideration. One way to modify our model to impose directionality is to specify edge directions between two nodes via some additional assumptions. For instance, in cases where two nodes are selected at each time step, the direction of the edge could be determined from the node with a lower weight or degree to the node with a higher weight or degree. In the future, it would be interesting to explore the effect of imposing network directionality on the network evolution (cf. ref. 28).

In spirit, our work conforms to the tradition of emphasizing the emergent scale-freeness of network evolution models. An interesting future direction would be to link this model to the other tradition of identifying scale-freeness by statistical tests[14]. One could potentially do this with a more direct statistical inference of the growth mechanisms (cf. ref. 44). Regardless, even in such a well-studied topic as general growth models for fat-tailed networks, there are open questions with unexplored solutions.

## Methods

### Degree and degree-ratio distributions of the nature–nurture model

Let node $i$ have weight $\omega_i$ and denote $n_i(T, k)$ as the probability that such a node has degree $k$ at time step $T$. Following standard process[3], we derive the Markovian rate equation for node $i$,

$$
\begin{aligned}
&n_i(T+1, k) \\
&= 2\Pi(\omega_i, k-1)n_i(T, k-1) + [1 - 2\Pi(\omega_i, k)]n_i(T, k),
\end{aligned}
\tag{8}
$$

where $\Pi(\omega, k)$ is the preferential probability given in the main text [Eq. (1)]. The initial condition of Eq. (8) is

$$
n_i(0, k) = \delta(k),
\tag{9}
$$

and the boundary condition is

$$
n_i(T, k) = 0, \text{ when } k < 0.
\tag{10}
$$

We are also interested in $P\big((k_i, k_j)|(i,j)\big)$, the conditional probability of randomly choosing a link that connects two nodes $i$ and $j$ of degrees $k_i$ and $k_j$, respectively. To avoid potential overcounting, we always call the first selected node as $i$ and the second selected node as $j$ in our bidirectional selection process, so that $(i,j)$ and $(j,i)$ are counted as different pairs by us. As a conditional probability, however, $P\big((k_i, k_j)|(i,j)\big)$ corresponds to the frequency of counting instances sampled from the pool of all links (~$T$), not nodes (~$N$), and therefore one cannot directly establish a Markovian rate equation that is similar to Eq. (8). To circumvent this, for any pair of nodes $i$ and $j$ with weights $\omega_i$ and $\omega_j$ respectively, we introduce an auxiliary variable $n_{ij}(T, k, k', (i,j))$ that denotes the joint probability of the spontaneous happening of three events at time step $T$: (1) node $i$ has degree $k$, (2) node $j$ has degree $k'$, and (3) $i$ and $j$ are connected.

Now, the Markovian rate equation for $n_{ij}(T, k, k', (i,j))$ is given by

$$
\begin{aligned}
&n_{ij}(T+1, k, k', (i,j)) \\
&= \Pi(\omega_i, k-1)\big[1 - \Pi(\omega_j, k')\big]n_{ij}(T, k-1, k', (i,j)) \\
&\quad + \big[1 - \Pi(\omega_i, k)\big]\Pi(\omega_j, k'-1)n_{ij}(T, k, k'-1, (i,j)) \\
&\quad + \big[1 - \Pi(\omega_i, k)\big]\big[1 - \Pi(\omega_j, k')\big]n_{ij}(T, k, k', (i,j)) \\
&\quad + \Pi(\omega_i, k-1)\Pi(\omega_j, k'-1)n_i(T, k-1)n_j(T, k'-1).
\end{aligned}
\tag{11}
$$

The first three terms of Eq. (11) account for the probability that, when nodes $i$ and $j$ are already connected at time step $T$, whether they will acquire (or not) a new link to satisfy the conditions on their degrees being $k$ and $k'$ at time step $T+1$. The last term accounts for the probability that, when nodes $i$ and $j$ are not connected at time step $T$ (which approximately happens with probability $n_i(T, k-1)n_j(T, k'-1)$ when the network is sparse), whether $i$ and $j$ will be connected and match all three conditions at the next time step. The initial condition of Eq. (11) is

$$
n_{ij}(0, k, k', (i,j)) = 0,
\tag{12}
$$

and the boundary conditions are

$$
n_{ij}(T, 0, k', (i,j)) = n_{ij}(T, k, 0, (i,j)) = 0.
\tag{13}
$$

If we can solve $n_i(T, k)$ [Eq. (8)] and $n_{ij}(T, k, k', (i,j))$ [Eq. (11)], which are functions of $\omega_i$ (and $\omega_j$), then both degree distribution and degree ratio distribution can be calculated given the node weight distribution $\rho(\omega)$, which we have assumed to be a continuous power-law distribution $\rho(\omega) = c\omega^{-\alpha}$ within the range $1 \le \omega \le \omega_{\max}$, given the normalization coefficient $c = (1-\alpha)/(\omega_{\max}^{1-\alpha} - 1)$. The DD is simply given by

$$
P(k) = \int_1^{\omega_{\max}} d\omega_i c\omega_i^{-\alpha}n_i(T, k).
\tag{14}
$$

To derive the degree ratio distribution, one has

$$
\begin{aligned}
&P\big((k_i, k_j), (i,j)\big) \\
&= \int_1^{\omega_{\max}} d\omega_i \int_1^{\omega_{\max}} d\omega_j c\omega_i^{-\alpha}c\omega_j^{-\alpha}n_{ij}(T, k_i, k_j, (i,j)),
\end{aligned}
\tag{15}
$$

which is the joint probability of randomly choosing a pair of nodes $i$ and $j$ that not only are connected but also have degrees $k_i$ and $k_j$. Then, the degree ratio distribution is given by[17]

$$
\begin{aligned}
P(\eta) &= \int_1^\infty 2k_i dk_i P\big((k_i, \eta k_i)|(i,j)\big) \\
&= \int_1^\infty \frac{2k_i N^2}{T} dk_i P\big((k_i, \eta k_i), (i,j)\big),
\end{aligned}
\tag{16}
$$

where in the second step we have used Bayes' rule, given that $P(i,j) = T/N^2$. Inserting Eq. (15) into Eq. (16) gives rise to $P(\eta)$.

Unfortunately, Eqs. (8) and (11) are difficult to solve. This is since the implicit time dependence of the preferential probability $\Pi(\omega, k)$ is intractable. However, special solutions can be found under certain limits:

1. In the nature-only limit, we can eliminate the nurture factor by letting $b \to \infty$, while keeping the power-law exponent $\alpha$ of the weight distribution being finite. This reduces Eq. (1) to

$$
\Pi(\omega, k) \simeq \frac{\omega}{N\bar{\omega}}, \text{ when } b \to \infty,
\tag{17}
$$

which is independent of $k$. Hence, our introduced model reduces to a pure bidirectional-selection fitness model with a power-law weight

(fitness) distribution, for which both the solutions of $P(k)$ and $P(\eta)$ are known[17]. The results are given in the main text [Eqs. (4) and (5)].

2. In the nurture-only limit, we can eliminate the nurture factor by letting $\alpha \to \infty$, which also implies $\omega_{max} \to 1$. This reduces Eq. (1) to

$$\Pi(\omega, k) \simeq \frac{k+b}{2T+bN}, \text{ when } \alpha \to \infty, \tag{18}$$

given that $\omega_i \simeq \bar{\omega} \simeq 1$ and $\rho(\omega) \simeq \delta(\omega - 1)$. Hence, our introduced model reduces to a preferential attachment model but without the growth of $N$. For small $b$, analytical solutions of $P(k)$ and $P(\eta)$ can be found [Eqs. (6) and (7)].

3. In the nature-nurture crossover, i.e., when both the bias $b$ and the power-law exponent $\alpha$ are finite, it is possible to derive an approximate solution by the following ansatz,

$$\sum_{i=1}^{N} \omega_i k_i \approx \chi \bar{\omega} T. \tag{19}$$

This is to explicate the time dependence of $\Pi(\omega, k)$ [Eq. (1)], by assuming that its denominator increases linearly with time $T$. Such a linear approximation is exact in the nurture-only limit (where $\chi = 2$), but we observe that the linear approximation still holds even when taking the nature factor into account, as long as the variance of $\omega$ is not too great. Since higher $\omega_i$ correlates with higher expectation of $k_i$, we expect the following inequality,

$$\sum_{i=1}^{N} \omega_i k_i \geq \sum_{i=1}^{N} \bar{\omega} k_i \geq 2\bar{\omega} T, \tag{20}$$

which implies $\chi \geq 2$. The more variability there is in the distribution of $\omega$, the larger $\chi$ is. To proceed, we employ numerical simulations to fix the parameter $\chi$. Specifically, given a set of model parameters $\omega_{max}$, $\alpha$, and $b$, we run simulations of the nature-nurture model and fit $\sum_{i=1}^{N} \omega_i k_i$ as a function of $T$, deriving the corresponding $\chi$. The parameter $\chi$ is further put in Eq. (1) to solve for $P(k)$ and $P(\eta)$, which, in turn, are used to fit the model parameters $\omega_{max}$, $\alpha$, and $b$. This leads to a set of self-consistent equations which converge to an optimal (or locally optimal) fit. For small $b$, the final solutions of $P(k)$ and $P(\eta)$ are similar to the nurture-only case, integrated over all possible $\omega_i$ and $\omega_j$, given by

$$P(k) \simeq \int_1^{\omega_{max}} d\omega_i c\omega_i^{-\alpha} e^{-B_i k} k^{A-1} B_i^A / \Gamma(A), \tag{21}$$

and

$$P(\eta) \simeq \int_1^{\omega_{max}} d\omega_i \int_1^{\omega_{max}} d\omega_j c\omega_i^{-\alpha} c\omega_j^{-\alpha}$$
$$2B_i^{A+1} B_j^{A+1} \eta^A E_{-2A-1}(B_j + \eta B_i) / \Gamma(A+1)^2, \tag{22}$$

where $A = b$ and $B_i = \left(b\chi^{-1} N T^{-1}\right)^{2\chi^{-1}\omega_i \bar{\omega}^{-1}}$, respectively. The analytical results agree with simulation results (Supplementary Fig. 1).

### Reporting summary
Further information on research design is available in the Nature Portfolio Reporting Summary linked to this article.

### Data availability
The processed data used in this study are available in the Colorado Index of Complex Networks (ICON) database [https://icon.colorado.edu].

### Code availability
Custom code that supports the findings of this study is available at https://github.com/bnzu/nnne.

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

## Acknowledgements

B.Z. was supported by the Startup Foundation for Introducing Talent of NUIST and the Qinglan Project of Jiangsu Universities. P.H. was supported by JSPS KAKENHI Grant Number JP 21H04595. Z.G. is supported by the National Natural Science Foundation of China with Grant Number 72371137. X.L. was supported by the National Nature Science Foundation of China with Grant Numbers 72025405, 72088101 and 72001211, and the Hunan Science and Technology Plan Project with Grant Number 2020TP1013.

## Author contributions

All authors contributed to the research. B.Z. and X.M. conceived the research, performed the experiments, and analyzed the data. Z.G., C.Z., and Y.H. cleaned the data. B.Z. and X.M. wrote the first draft of the manuscript. P.H. and X.L. reviewed and edited the manuscript.

## Competing interests

The authors declare no competing interests.
