## [Peer Review File · Nature Communications]

REVIEWER COMMENTS

Reviewer #1 (Remarks to the Author):

This work is exciting, easy to read and is built on a 2-parameter model only, yet providing insights into long-standing controversies in network science: Are the node degrees genuine power laws in many (social) networks, if they would grow ad infinitum?

(Finite networks cannot be slowly decaying power laws).

What is the minimal model to describe the (broad) tail distribution of the node degree?

What is the minimal model to describe the bulk of the degrees (the smaller ones)?

What is the minimal model to describe both regimes? Ultimately: Is there any model?

The authors have provided possible answers to all those long standing questions in network science by a model that shows R^2 fits around 99% for the most important (social) network data we have. Whereas the node importance (a big shot professor like me, a large airport hub) has been a bit behind the scenes in network growth models, the present study emphasises its importance -- if combined with the link's weights. In addition, the authors provide a plausible interpretation for those two fundamental effects: nurture and nature. Equally important, the model explains also (for some networks), to certain degree, their evolutionary process -- given that the main parameters, "degree cutoff" and the b -parameter vary (or stay constant) accordingly.

Taken together, this is one of the simplest,

robust and most convincing models that I have seen in years in network science, and it is beautiful because it is so beautifully parsimonious and elegant, yet revealing a combination of two mechanisms that nobody has seen before in the field, yet -- even if one can say that many publications were pretty close to this (and the effects have been studied previously, in some ways, but separately).

I recommend publication and a highlight in Nature Communication on this.

minor points:

sorrel node: rephrase or explain,
not all readers are familiar with sorrel.

in the paper, a node's strength is not discussed,
which would also be a measure of a node's importance of some sort
as it usually refers to the sum of its weights of outgoing/incoming links.
it is ok to invent a new name for a new thing but the "nature" notion
could be better embedded in the literature.

AICc: is there any ref given?

if not: what about the book of
Burnham and Anderson?

Reviewer #2 (Remarks to the Author):

The authors propose an elegant and simple model that allows disentangling two critical aspects in the network evolution: the weight of the nodes (which can be perceived as their intrinsic attractiveness) and their degree (which relates to their luck in accumulating connections).

Understanding the evolution of complex systems, e.g., social systems, remains an essential topic in the nowadays inter-connected world. The work is embedded in the rich literature on the topic, and the results are presented clearly.

However, I have a number of suggestions for improving the manuscript:

1. The work focuses on two aspects: the weight and the degree of the nodes. While the degree is a clearly quantifiable object, e.g., the number of connections in a social network, an in-depth discussion on the definition of the nodes' weight is currently missing. Relate to this, I have a number of suggestions:

- a. an intuition behind the meaning of the nodes' weight should be already expressed in the abstract or, at the very least, in the introduction (when it is first mentioned).

b. the authors should also compare their definition of weight with that of fitness (Caldarelli, Guido, et al. "Scale-free networks from varying vertex intrinsic fitness." Physical review letters 89.25 (2002): 258702.) and that of quality (Pagan, Nicolò, et al. "A meritocratic network formation model for the rise of social media influencers." Nature communications 12.1 (2021): 6865.)

c. the results (e.g., eq (5)) seem to heavily rely on the assumption that the weight follows a power-law distribution. To support this assumption, they argue that wealth is power-law distributed. However, wealth can be the result of some underlying process, and uniform quality (weight) distribution can still lead to power-law effects (see again Pagan et al. 2021). What happens to the model/results if the assumption does not hold?

2. Regarding the model: while I believe it is reasonable to find a trade-off between the role of luck (accumulated degree) and the intrinsic attractiveness of the nodes, I think the authors should better motivate why it is reasonable to assume that Equation (1) provides the right way of modeling this interplay.

Furthermore, when they mention the snowball effect for nurture attractiveness: isn't it due to the effect of the recommender systems (see Pagan et al. 2021)?

3. Regarding the results: the authors validate the model by finding the best parameter fitting to some real-world networks.

a. First, authors could elaborate more on their method to find the best optimal fit. Furthermore, it would be interesting to have an estimate of possible confidence intervals. Namely, it would be interesting to see if the network formation process is less sensitive to some of these parameters. A big part of the claim on the nature-nurture origin of social vs non-social networks depends on this.

b. In Fig 3. why are the authors plotting β ? What about α (from the table in the SI, it appears there is a much larger variability in the estimates of α)? Furthermore, their claim on "nature"-driving model is for $\beta \rightarrow \infty$. Here, the estimates of β are between 0 and 2, i.e., not really large values, especially compared to the degrees. I think plotting β is kind of misleading, and the main difference between the estimates concerns w_{\max} , i.e., the underlying weight distribution, which seems to indicate that in social networks weights are more concentrated (unif. distributed) than in non-social networks.

c. Related to the above point: an interesting analysis would be to look at the different estimated weight distributions for the different networks.

d. Authors are working with directed networks. However, the network formation/evolution process for directed and undirected networks can be very different. Therefore, I think the authors should comment on why they can reasonably neglect this difference. Furthermore, I think some of the networks, e.g., word adjacency networks, are a bit out-of-scope.

4. In order to provide validation to their methodology, the authors show that calibrating their model with the best fitting parameters allows them to obtain a close match in the degree distribution as well as in the degree-degree ratio distribution. However, it remains unclear whether:

a. the model is overfitting, i.e., it has a sufficiently large number of parameters (currently 3) that it can accurately fit any arbitrary distribution.

b. the best-fitted parameters are somehow realistic for the related networks.

b. the model produces other realistic network features other than the two aforementioned.

Minor comments:

1. Table 3 (in the SI) could be converted into a colormap to aid in visualizing the results.

2. the order of the networks could be changed so that the social networks are all at the top, and the non-social networks are all at the bottom (or vice-versa) since many claims relate to this distinction.

3. in Line 59: is "index" referring to the index of the power-law distribution?

4. in Line 189: social eleven  eleven social

Our replies to the reviewers' comments follow below.

Reviewer #1

Thank you very much for your feedback. It has been a journey of ten years, from the inception of this project in 2013 to its final completion. Praise and recognition like your report makes the effort worthwhile.

sorrel node: rephrase or explain,
not all readers are familiar with sorrel.

We have done it in the revised manuscript.

in the paper, a node's strength is not discussed,
which would also be a measure of a node's importance of some sort
as it usually refers to the sum of its weights of outgoing/incoming links.

This is an excellent question. The issue of node strength has been considered in many studies, typically referring to the sum of weights of all links of a node. In our work, our primary focus is to explore the evolutionary origins of real-world complex networks from a macro-philosophical framework encompassing the nature and nurture factors.

|

As a result, we aimed to develop a concise yet universally applicable model to unveil the roots of complex network evolution. After years of exploration and experimentation, we found that the nature-nurture model proposed in this paper achieves this goal. Within the simulated networks generated by the nature-nurture model, statistically speaking, nodes with larger weights tend to have higher degrees. Therefore, we consider that the weight representing the nature factor of nodes in the nature-nurture model bears similarity to the concept of node strength commonly discussed in prior research.

it is ok to invent a new name for a new thing but the "nature" notion could be better embedded in the literature.

We have added references to the literature to clarify our usage of “nature.” The references are as follows.

Caldarelli, G., Capocci, A., De Los Rios, P. & Munoz, M. A. Scale-free networks from varying vertex intrinsic fitness. *Phys. Rev. Lett.* 89, 258702 (2002)

Pagan, N., Mei, W., Li, C. & Dörfler, F. A meritocratic network formation model for the rise of social media influencers. *Nat. Commun.* 12, 6865 (2021).

AICc: is there any ref given?
if not: what about the book of
Burnham and Anderson?

We are now citing Burnham and Andersson and give the details of how we calculate AIC.

Reviewer #2

Thank you very much for your recognition of our work and the valuable suggestions you've provided for revisions. Following your feedback, we have made our best efforts

to revise the manuscript according to your comments. We believe the manuscript's quality has truly increased.

1. The work focuses on two aspects: the weight and the degree of the nodes. While the degree is a clearly quantifiable object, e.g., the number of connections in a social network, an in-depth discussion on the definition of the nodes' weight is currently missing. Relate to this, I have a number of suggestions:

a. an intuition behind the meaning of the nodes' weight should be already expressed in the abstract or, at the very least, in the introduction (when it is first mentioned).

We have included an explanation of node weights in the abstract, signifying the hierarchical attributes of nodes within the network.

b. the authors should also compare their definition of weight with that of fitness (Caldarelli, Guido, et al. "Scale-free networks from varying vertex intrinsic fitness." *Physical review letters* 89.25 (2002): 258702.) and that of quality (Pagan, Nicolò, et al. "A meritocratic network formation model for the rise of social media influencers." *Nature communications* 12.1 (2021): 6865.)

Both references are very important works that focus on node weights/fitness. These references are now added.

c. the results (e.g., eq (5)) seem to heavily rely on the assumption that the weight follows a power-law distribution. To support this assumption, they argue that wealth is power-law distributed. However, wealth can be the result of some underlying process, and uniform quality (weight) distribution can still lead to power-law effects (see again Pagan et al. 2021). What happens to the model/results if the assumption does not hold?

This is an excellent suggestion. we notice that mathematical form of the power-law

weight distribution ($\sim w^{-\alpha}$) also covers cases of a uniform distribution when $\alpha=0$, or a short-tail (e.g., exponential) distribution when $\alpha \rightarrow \infty$. To check the possibility of a uniform distribution, we present the AICc statistical results for the optimal fitting of the degree distribution and degree-degree ratio distribution of the nature-nurture model with $\alpha = 0$ (uniform), across thirty-two real networks in SI Table 3 and SI Fig. 4. By comparing the nature-nurture model (α free) and the nature-nurture model with $\alpha=0$ (fixed), we find that the nature-nurture model (α free) is the most favored by AICc for all thirty-two real networks except for one network. This indicates the necessity of the assumption that node weights in the nature-nurture model adhere to a power-law (or power-law-like) distribution, which plays a significant role in the evolution of networks.

2. Regarding the model: while I believe it is reasonable to find a trade-off between the role of luck (accumulated degree) and the intrinsic attractiveness of the nodes, I think the authors should better motivate why it is reasonable to assume that Equation (1) provides the right way of modeling this interplay.

This suggestion is greatly appreciated. Our choice of Eq. (1) is inspired by the Bianconi–Barabási model, where k and w are also combined multiplicatively, contributing to the preferential probability P of a node being selected. In our model, a new constant \max is added. This constant, indeed, represents the major discovery of our work, as it faithfully determines the relative contributions of the nature/nurture factors.

Furthermore, when they mention the snowball effect for nurture attractiveness: isn't it due to the effect of the recommender systems (see Pagan et al. 2021)?

That is a great point. We mention it along with the citation to Pagan et al.

0. Regarding the results: the authors validate the model by finding the best parameter fitting to some real-world networks.

a. First, authors could elaborate more on their method to find the best optimal fit. Furthermore, it would be interesting to have an estimate of possible confidence intervals. Namely, it would be interesting to see if the network formation process is less sensitive to some of these parameters. A big part of the claim on the nature-nurture origin of social vs non-social networks depends on this.

This is an excellent suggestion. In the revised manuscript's Appendix and SI, we have provided an explanation of our methodology for determining the optimal fitting parameters for degree distribution and degree-degree ratio distribution. Additionally, we have included the confidence intervals for these optimal parameters.

b. In Fig 3. why are the authors plotting β ? What about α (from the table in the SI, it appears there is a much larger variability in the estimates of α)? Furthermore, their claim on "nature"-driving model is for $\beta \rightarrow \infty$. Here, the estimates of β are between 0 and 2, i.e., not really large values, especially compared to the degrees. I think plotting β is kind of misleading, and the main difference between the estimates concerns w_{\max} , i.e., the underlying weight distribution, which seems to indicate that in social networks weights are more concentrated (unif. distributed) than in non-social networks.

These are all fair points. We have examined the distribution of three parameters, namely β , w_{\max} , and α , within social and non-social networks, revealing that only w_{\max} exhibits distinct distribution ranges across these networks. In Fig. 3 (a), we have redrawn the distribution of w_{\max} within social and non-social networks.

c. Related to the above point: an interesting analysis would be to look at the different estimated weight distributions for the different networks.

We have shown the dependence between β and w_{\max} (see the figure below). In general, the smaller w_{\max} is, the larger β seems to be, hinting that the distribution is

more concentrated. However, α does not show the same level of distinction of social vs. non-social networks as O_{\max} shows. Therefore, we choose O_{\max} as the sole indicator of nature/nurture comparison.

d. Authors are working with directed networks. However, the network formation/evolution process for directed and undirected networks can be very different. Therefore, I think the authors should comment on why they can reasonably neglect this difference. Furthermore, I think some of the networks, e.g., word adjacency networks, are a bit out-of-scope.

This is an excellent suggestion. In the final section of the revised manuscript, we have added our discussion:

In our work, we have not explicitly addressed the issue of network directionality. The primary goal of our study is to investigate the universal mechanisms that can be adaptable to the evolution of both undirected and directed networks. For empirical

studies involving directed networks, we treat the sum of node outdegrees and indegrees as the total degrees of a node, followed by calculating the degree distribution without explicitly delving into the directionality consideration.

One way to modify our model to impose directionality is to specify edge directions between two nodes via some additional assumptions. For instance, in cases where two nodes are selected at each time step, the direction of the edge could be determined from the node with a lower weight or degree to the node with a higher weight or degree. In the future, it would be interesting to explore the affect of imposed network directionality on the network evolution (as in Pagan et al. 2021).

Meanwhile, we disagree that the word-adjacency networks fall without the scope of our model's explanation. One could indeed design a more specific model for this particular case, but at the very coarse level of abstraction that our model operates at, covering very general network growth processes, the conclusion that word-adjacency networks are closer to "nature" than "nurture" should be valid.

4. In order to provide validation to their methodology, the authors show that calibrating their model with the best fitting parameters allows them to obtain a close match in the degree distribution as well as in the degree-degree ratio distribution. However, it remains unclear whether:
 - a. the model is overfitting, i.e., it has a sufficiently large number of parameters (currently 3) that it can accurately fit any arbitrary distribution.

Despite the presence of three parameters in the nature-nurture model, we have presented the AICc statistical results for optimal fitting across thirty-two real networks for various model configurations, including the nature-nurture model (three parameters), nature model (two parameters), and nurture model (one parameter). The Akaike information criterion (AIC) is an estimator of prediction error and thereby relative quality of statistical models for a given set of data. AIC deals with the trade-off between the goodness of fit of the model and the simplicity of the model. In other words, AIC deals

with both the risk of overfitting and the risk of underfitting. By comparing the nature-nurture model, the nature model, the nurture model, we find that the nature-nurture model is the most favored by AICc for all thirty-two real networks except for one network, suggesting that the nature-nurture model is not overfitting.

b. the best-fitted parameters are somehow realistic for the related networks.

Regarding the rationality of the optimal fitting parameters of the nature-nurture model for the thirty-two real networks, our model does not impose any specific regulations or constraints on the ranges of the three fitting parameters: α , W_{\max} , and b , apart from the requirement that all three parameters being greater than 0. We infer that your concern likely pertains to the rationality of the optimal fitting results for α . As α serves as the exponent for the weight distribution, its range is typically between 0 and 20. However, extensive empirical studies suggest that the exponent of degree distribution in real networks generally falls within the range of [1, 5]. Yet, it is important to distinguish between the exponent of weight distribution in the nature-nurture model and the exponent of degree distribution in real networks. Indeed, the weight distribution may not be power law *a priori*. Our proposed mathematical form of the power-law weight distribution ($\sim W^{-\alpha}$) covers not only power laws, but also cases of a uniform distribution when $\alpha=0$, or a short-tail (e.g., exponential) distribution when $\alpha \rightarrow \infty$. Therefore, a realistic explanation of our finding of a large α (~ 20) is not that the weight distribution follows a precise, very-sharp power law, but rather it follows some short-tailed distribution, for example, an exponential distribution ($\alpha \rightarrow \infty$). This consideration is discussed in the SI.

d. the model produces other realistic network features other than the two aforementioned.

Your feedback is highly valuable, and incorporating additional statistical metrics would indeed offer a more comprehensive validation of the model's scientific soundness and

rationality. We recognize that obtaining mathematical analytical solutions for the degree distribution and degree-degree ratio distribution within the nature-nurture model has proven to be highly challenging, requiring significant dedication. On one hand, identifying another suitable network statistical metric akin to the degree distribution and degree-degree ratio distribution is not straightforward. On the other hand, even if we were to identify alternative suitable statistical metrics, it's possible that their solution processes could be even more complex than that of degree distribution and degree-degree ratio distribution. We remain uncertain about the feasibility of deriving solvable solutions for such metrics.

Minor comments:

1. Table 3 (in the SI) could be converted into a colormap to aid in visualizing the results.

We tried a colormap, but it was not as informative as a table. As a remedy, we adjusted the table by shifting each AICc by a constant (because the absolute value of AICc does not have meanings), so that for the nature-nurture model it is always zero. This allows a much easier way to compare with the AICc (also shifted) of other models.

2. the order of the networks could be changed so that the social networks are all at the top, and the non-social networks are all at the bottom (or vice-versa) since many claims relate to this distinction.

In the revised version, we have reorganized all the real networks listed in Table 1 into separate categories of social and non-social networks.

3. in Line 59: is "index" referring to the index of the power-law distribution?

Your understanding is correct. In the phrase "rather than a specific index," the term 'index' refers to statistical indicators such as power-law distribution for degree

distribution. We believe that interpreting the scale-free property of networks as a consequence of a certain mechanism, like preferential attachment, rather than determining whether a network possesses this property based on a specific statistical index, offers a more universally applicable criterion. This approach also has the potential to address the longstanding debate regarding whether complex networks exhibit scale-free properties.

4. in Line 189: social eleven  eleven social

In the revised version, we have corrected this clerical error.

REVIEWERS' COMMENTS

Reviewer #2 (Remarks to the Author):

I truly appreciate the authors' engagement with the proposed suggestions, which ultimately led to an improved manuscript version. Hence, I consider it ready for publication.

Dear editors and reviewers,

Because the reviewers had no further revision comments and agreed to publish our paper, we did not make any major revisions to our manuscript, but only made minor adjustments to the format according to the requirements of the journal.